# BotCl, the First Chlorotoxin-like Peptide Inhibiting Newcastle Disease Virus: The Emergence of a New Scorpion Venom AMPs Family

**DOI:** 10.3390/molecules28114355

**Published:** 2023-05-26

**Authors:** Abir Jlassi, Marwa Mekni-Toujani, Asma Ferchichi, Charfeddine Gharsallah, Christian Malosse, Julia Chamot-Rooke, Mohamed ElAyeb, Abdeljelil Ghram, Najet Srairi-Abid, Salma Daoud

**Affiliations:** 1LR20IPT01 Laboratoire des Biomolécules, Venins et Applications Théranostiques (LBVAT), Institut Pasteur de Tunis, Université de Tunis El Manar, Tunis 1002, Tunisia; 2LR16IPT03 Laboratoire d’Epidémiologie et MicrobiologieVétérinaire, Institut Pasteur de Tunis, Université de Tunis El Manar, Tunis 1002, Tunisia; 3LR16IPT02 Laboratoire de Recherche sur la Transmission, le Contrôle et l’Immunobiologie des Infections, Institut Pasteur de Tunis, Université de Tunis El Manar, Tunis 1002, Tunisia; 4Mass Spectrometry for Biology Unit, Institut Pasteur, Université Paris Cité, CNRS UAR 2024, 75015 Paris, France

**Keywords:** *Buthus occitaus tunetanus* venom, chlorotoxin-like peptides “BotCl”, antiviral activity, Newcastle disease virus

## Abstract

Newcastle disease virus (NDV) is one of the most serious contagions affecting domestic poultry and other avian species. It causes high morbidity and mortality, resulting in huge economic losses to the poultry industry worldwide. Despite vaccination, NDV outbreaks increase the need for alternative prevention and control means. In this study, we have screened fractions of *Buthus occitanus tunetanus* (*Bot*) scorpion venom and isolated the first scorpion peptide inhibiting the NDV multiplication. It showed a dose dependent effect on NDV growth in vitro, with an IC_50_ of 0.69 µM, and a low cytotoxicity on cultured Vero cells (CC50 > 55 µM). Furthermore, tests carried out in specific pathogen-free embryonated chicken eggs demonstrated that the isolated peptide has a protective effect on chicken embryos against NDV, and reduced by 73% the virus titer in allantoic fluid. The N-terminal sequence, as well as the number of cysteine residues of the isolated peptide, showed that it belongs to the scorpion venom Chlorotoxin-like peptides family, which led us to designate it “BotCl”. Interestingly, at 10 µg/mL, BotCl showed an inhibiting effect three times higher than its analogue AaCtx, from *Androctonus australis* (Aa) scorpion venom, on NDV development. Altogether, our results highlight the chlorotoxin-like peptides as a new scorpion venom AMPs family.

## 1. Introduction

Newcastle disease (ND) is a highly contagious viral disease affecting many bird species. It is one of the most important and serious infections for the poultry industry, causing severe economic losses worldwide [1,2]. The causative agent of this serious infection is the Newcastle disease virus (NDV), a virulent form of Avian Paramyxovirus serotype1 (AMPV-1) belonging to the genus *Avulavirus*, the family *Paramyxo viridae* and the order *Mononega virales* [3]. It is an enveloped virus containing a single stranded, non-segmented, negative sense RNA genome, which encodes six structural proteins consisting of the nucleoprotein (NP), the phosphoprotein (P), the matrix protein (M), the fusion protein (F), the hemagglutinin-neuraminidase (HN) and the RNA polymerase (L) [4].

The HN has three functions: (i) it recognizes sialic-acid-containing receptors on cell surfaces, (ii) promotes the fusion activity of the F protein, allowing the virus to penetrate the cell surface and (iii) acts as a neuraminidase (sialidase), removing sialic acids from progeny virus particles to prevent viral self-agglutination, thereby aiding viral spread [3].

Vaccination is the current means for the control and prevention of NDV infection [5]. However, despite the availability of various types of vaccines (live, inactivated and recombinant vaccines), NDV outbreaks are still occurring due to a lack of efficacious sanitary protocols and especially to mutations within viral strains, resulting in significant differences between current vaccine strains and field prevailing ones [6,7]. On the other hand, anti-NDV drugs are not available, due to their significant toxicity and high production cost [8]. Therefore, the development of efficient antiviral molecules is crucial to fight NDV infections. Natural resources, such as animal venoms, have been shown to be endowed with anti-microbial effects [9].

In particular, scorpion venom is a complex mixture containing neurotoxins, protease inhibitors, enzymes, antimicrobial peptides (AMPs) and some other compounds [10,11]. AMPs from scorpion venoms have demonstrated relevant activities against bacteria, fungi, and viruses [11].

Scorpion venom AMPs are positively charged amphipathic peptides and can be conveniently divided into three structural categories: (1) cysteine containing peptides with disulfide bridges, called disulfide-bridged (DB) AMPs; (2) peptides with an amphipathic α-helix but lacking cysteine residues, called non-disulfide-bridged (NDB) AMPs and (3) peptides rich in certain amino acids, such as proline and glycine [10].

More attention is being given to antiviral peptides, due to their unique biological activities that can make them potentially useful as antiviral drugs [12]. Therefore, these studies highlight the diversity and the complexity of such arachnid venoms, which need to be explored for new antiviral candidates.

In this study, we screened *Buthus occitanus tunetanus* (*Bot*) scorpion venom fractions for their anti-NDV effect and identified a new peptide exhibiting in vitro and in ovo antiviral activities. It appears that this peptide belongs to a scorpion venom peptides family, not reported to exhibit a antimicrobial effect before now.

## 2. Results

### 2.1. Screening for Anti-NDV Active Molecules from Bot Scorpion Venom

To search for fraction/peptide(s) able to inhibit NDV infection, the crude venom of *Bot* scorpion was fractionated by gel filtration with Sephadex G50 column chromatography. Four fractions, M1, M2, BotG50 and M3, were eluted [13]. The BotG50 fraction was known to contain peptides active on ion channels and endowed with pharmacological activities [14,15,16]. Accordingly, this fraction was selected for further purification. Indeed, this fraction still contains a huge number of proteins/peptides, as demonstrated by the SDS-PAGE in Figure 1A. This figure showed that, at the same concentration of 50 µg, the crude venom (Bot) and the BotG50 fraction contain roughly the same protein profile. The BotG50 fraction contained a heterogeneous mixture of molecules of different molecular mass, while it was expected that it would contain fractions with nearly the same molecular mass (MM) since the separation process is mainly based on MM. However, this process involves an ideal mechanism where the analytes have no attractive or repellent interaction with column filling, except for the effects caused by the impermeability of the pore walls, and the dependence between the molecular size and the MM is applicable only for molecules with similar structures.

The fast protein liquid chromatography (FPLC) of the BotG50 fraction on a cation exchange Resource S column allowed different fractions separated according to their net charge (Figure 1B). All these fractions were tested, at 10 µg/mL, on Vero cells against NDV infectivity. This concentration was selected as it represented a significant effect in the previous study [17]. Only fractions F13, F16 and F18 are able to inhibit NDV infection. The F16 fraction, eluted at 42 min, exhibited the highest inhibitory activity (83%) against NDV infectivity, as compared when tested to F13 (38%) and F18 (49.33%) (Figure 1C).

The fraction F16 was then injected into an RP-C18 high performance liquid chromatography (HPLC) column for purification. The fraction corresponding to the peak number 6 (H6), eluted at 26 min (Figure 1D), showed the most significant antiviral activity, with 78.41% inhibition at 10 µg/mL concentration (Figure 1E).

This fraction represented 0.02% of the crude *Bot* venom.

### 2.2. Mass Spectrometry and Sequence Analysis

Mass spectrometry analysis of H6 showed a compound with three to five charge states, giving a mass of 3625.4071 Da after the spectra deconvolution (Figure 2A). After reduction with TCEP, the mass of the peptide showed an increase of 8.005 Da (Figure 2A). Owing to a mass increase of 2 Da per disulfide reduction [18], our results revealed the presence of four disulfide bridges in the structure of the peptide. The HCD MS/MS spectrum was recorded from the 4^+^ charge state ion 909.87 Da obtained after TCEP reduction, allowing the identification of the first 17 amino acids of the peptide sequence. Due to its very low concentration in the venom, we could not determine the complete amino-acid sequence of this peptide. Nevertheless, the analysis of the N-terminal sequence of this peptide (Figure 2B) showed a high similarity with those of the chlorotoxin-like peptide family represented by Chlorotoxin (CTX) (69.23% identity) [19], a toxin active on the chloride channel from *Leiurus quiquestriatus* scorpion venom. The highest similarity was found with AaCtx (87% identity), from *Androctonus australis* (Aa) scorpion venom [20], and with the insect-toxin, I5A (84.62% identity) [21], from *Buthus eupeus* scorpion venom (Figure 2C). Consequently, we named this peptide BotCl.

### 2.3. BotCl Exerts In Vitro a Potent Dose–Response-Specific Effect on NDV Multiplication

To determine whether BotCl is appropriate for further development as an anti-NDV candidate, its effect on Vero cells’ viability was assessed by MTT assay. Vero cell layers were incubated with serial dilutions of BotCl, ranging from 100 µg/mL to 5 µg/mL. After 72 h incubation, the peptide did not show a significant effect on Vero cells’ viability. Thus, the 50% cytotoxicity concentration (CC_50_) of BotCl was >100 µg/mL (27.6 µM) (Figure 3A).

Subsequently, different concentrations of BotCl (10 µg/mL; 5 µg/mL; 2.5 µg/mL; and 1.25 µg/mL) were incubated with NDV (MOI = 0.001) and then inoculated on Vero cells. Our results showed that BotCl exerted a dose-dependent antiviral effect with an IC_50_ value of 2.5 µg/mL (0.69 µM) (Figure 3B). The selectivity index of BotCl was thus >40, indicating a good selectivity of BotCl against NDV. The concentration of 10 µg/mL (2.76 µM) was chosen for further anti-NDV experiments, as it was not cytotoxic to Vero cells and demonstrated a significant anti-NDV effect.

Since the N-terminal sequence of BotCl showed the highest similarity with AaCtx, we compared the effects of the two peptides (at a concentration of 10 µg/mL) on the NDV multiplication. Our result revealed that BotCl displayed about three times higher activity, with 81.5% inhibition compared to AaCtx, which inhibited 32.22% of the NDV multiplication at the same concentration (Figure 4).

### 2.4. In Vitro NDV Inhibitory Mechanisms of BotCl Peptide

To gain insights into the effect of BotCl on the life cycle of NDV, different assays were performed, as described in Section 2.7 (Figure 5A). We found that when BotCl was added two hours before viral infection, it did not show any inhibitory effect. Similarly, when applied to infected Vero cells, BotCl failed to inhibit the replication of NDV into the host cells, demonstrating its ineffectiveness as an antiviral agent at this stage (Figure 5B).

However, when BotCl and the virus were mixed prior to being added to the Vero cells, and incubated for two hours, we observed a high inhibition of viral multiplication (Figure 5B). The inhibition decreased when both BotCl and NDV were added directly and simultaneously to Vero cells. These results advocate a direct interaction of BotCl with NDV virus. This interesting antiviral property prompted us to evaluate the antiviral activity of BotCl in ovo.

### 2.5. BotCl Did Not Induce In Ovo Toxicity

BotCl was tested at different concentrations (6.25; 12.5; 25; 50; 100 µg/mL) for its effect on 9-day-old SPF embryonated eggs. After 7 days’ incubation, it appeared that the size and the general aspect of treated chicken embryos were comparable to those of untreated ones, contrary to the BotG50 fraction, which showed a high hemorrhagic effect after only 2 days’ incubation (Figure 6). Furthermore, no macroscopic lesions wereobserved on the live chicken embryos treated with BotCl. These results showed that up to a dose of 100 µg/mL (27.6 µM) BotCl did not show any toxic effect on the embryos, and could be used for anti-NDV in ovo assays.

### 2.6. BotCl Protected Chicken Embryos against NDV

When BotCl (10 µg/mL), was pre-incubated with NDV, for 2 h at 37 °C, and inoculated into SPF embryonated eggs, a decrease in the size of the embryos by approximately 25%, compared to the untreated chicken embryos (NC), was detected. However, the embryos inoculated by the NDV alone were dead after 3 days of the virus infection (Figure 7A). Thus, the results showed that BotCl, at 10 µg/mL, exhibits a partial inhibitory effect of NDV infectivity and presents a protective effect on chicken embryos against NDV infection.

Furthermore, the NDV titers in BotCl-treated and untreated egg embryos measured in the collected allantoic fluids were 10^2.3^ EID_50_/mL and 10^8.5^ EID_50_/mL, respectively, demonstrating that the scorpion venom peptide BotCl was able to inhibit virus replication in ovo by 73% (Figure 7B).

### 2.7. A Putative New Family of Scorpion Venom AMPs

#### 2.7.1. Sequence Alignment and Phylogenetic Study

Since both the chlorotoxin-like peptides BotCl and AaCtx presented an antiviral effect on NDV, we assumed that other chlorotoxin-like peptides may have antimicrobial effects. Thus, we studied the sequence homology of the known Chlorotoxin-like peptides compared with those of the disulfide bridged antimicrobial peptides from scorpion venoms, based on similar activities emerging from a structure homology.

As shown in Figure 8A, the sequences of chlorotoxin-like peptides can be well aligned with those of scorpion venom DB-AMPs. Particularly, the Cys16, Cys20, Cys28, Cys33 and Cys35 of ClTx occupy conserved positions in all the sequences. Other amino acids were also conserved in the majority of the peptides such as Gly30 (in the ClTx sequence), as well as Gly26, which was conserved in all the sequences except for BeI5.

In the context of supposing the presence of an evolutionary relationship between all scorpion venom AMPs, we also introduced the sequences of the NDB-AMPs. All the sequences were aligned and used to establish the phylogenetic tree, according to the maximum likelihood method JTT model (Figure 8B). As shown in Figure 8B, five major clades emerged from the phylogenetic tree, distributed mainly according to the type of AMPs, thus presenting cladistic and evolutionary relationships between the NDB- and DB-AMPs. The first clade comprised similar sequences of NDB-AMPs, including Hp, VmCT and StCT. Peptides from *Pandinus cavimanus* scorpion venom are also included in this clade. The second clade also contained NDB-AMPs, including Opistoporin, Vejovine, Heterin-1, BmKbpp and Opistoporin. Clade III and Clade IV revealed relationships between NDB- and DB-AMPs, as they included Meucin-13 (Clade III) and OpiScorpin (Clade IV) as NDB-AMPs with HgebKTx (Clade III), EV37, Smp76 and BmKDfsin (Clade IV) as DB-AMPs. Interestingly, at the level of Clade V, we noted a cladistic relationship between different types of AMPs, particularly regarding the chlorotoxin-like peptides, which interlaced with NDB-AMPs (including antibacterials TsAP-1 and TsAP-2 as well as antivirals Mucroporin-M1 and Ctry2801).

Thus, all these results converge towards the birth of a new family of chlorotoxin-type AMPs, structurally related to those already reported.

#### 2.7.2. Study of the AaCtx with HN-NDV Interaction by Molecular Docking

The results of the molecular docking between AaCtx and HN-NDV generated 100 solutions ranked according to their docking score. The top ten ranked solutions were retained for a visual examination. Four of these solutions (second, third, fifth and sixth) bound to the previously identified binding site of HN [22]. These had different free binding energies and dissociation constants (Kd)at 37 °C (Appendix A). The third solution, showing the lowest binding energy (−13.3) and Kd (4.1 × 10^−10^), was selected.

As shown in Figure 9A, the AaCtx inserted into the binding site cavity of the NDV glycoprotein. It bound mostly with its loop linking the α-helix to the β-sheet, mainly by G21, R23 and R24 (Figure 9A). These residues interacted with R416, E401 and E258, respectively (Figure 9B). In addition, the residue S22 interacted with R416, E401 and Y526.

## 3. Materials and Methods

### 3.1. Chemicals and Reagents

The *Bot* scorpions were collected from Beni Khedach (South of Tunisia) by the veterinary services of the Institute Pasteur of Tunis. The venom was obtained through electric stimulation of scorpion post-abdomen and kept frozen at −20 °C, in its crude form, until used. Chemicals (reagent grade) were purchased from Sigma–Aldrich^®^ (St. Louis, MO, USA) chemical company, except otherwise indicated. African green monkey kidney cells (Vero, ATCC CCL 85) were purchased from the American Type Culture Collection (ATCC, Boston, MA, USA) and the live virus vaccine ND (LaSota strain) (Cevac New L, Budapest, Hongrie) was used in all the assays. Cell culture supplements and reagents were purchased from GIBCO (Cergy-Pontoise, France); UltraPure™ Agarose was from Invitrogen (Waltham, MA, USA).

### 3.2. Purification

Crude venom (250 mg) from *Bot* scorpion was dissolved in cold water (1:4 *v*/*v*), and centrifuged at 15,000× *g* for 15 min. The supernatant was loaded on Sephadex G-50 gel filtration chromatography column (K26/100), equilibrated at room temperature (25 °C) and eluted with 0.1 M acetic acid at a flow rate of 0.5 mL/min. The collection was carried out by an automatic fraction collector every 5 min with a volume collection of 2.5 mL per tube. The fractions pooling led to the obtaining of 4 fractions, as previously described [23]. Only the fraction named BotG50, containing toxins with molecular mass ranging between 3000 Da and 7000 Da, was retained.

After freeze-drying, BotG50 was fractionated using FPLC on a cation exchange Resource S column (GE Healthcare, Uppsala, Sweden), pre-equilibrated by a solution buffer of 0.05 M ammonium acetate (pH 6.7). Molecules were eluted at a constant flow rate of 1 mL/min, using a linear gradient from 0.05 to 0.5 M ammonium acetate, for 58 min. Absorbance was monitored at 280 nm and fractions were collected manually and lyophilized, then dissolved in water. The obtained fractions were tested for their activities against NDV development. The fraction showing the best antiviral activity was purified on a C18 reversed phase HPLC column (250 × 10 mm, 5 μm; Beckman Fullerton, CA, USA), equilibrated in 0.1% trifluoro-acetic acid (TFA) in water. Peptides were eluted at a flow rate of 0.8 mL/min, using a linear gradient from 10 to 100% of solvent B (0.1% TFA in acetonitrile) in solvent A (0.1% TFA in water) in 45 min and collected according to their absorbance, monitored at 214 nm. Peptide concentration was determined by a BCA assay kit (Sigma-Aldrich, Burghausen, Germany), using Bovine Serum Albumin as a standard.

### 3.3. Mass Spectrometry and Amino Acid Sequence Determination

The H6 fraction was desalted by C18 ZipTip^®^ (Millipore, Taufkirchen, Germany) and eluted directly into a 10 µL spray solution of acetonitrile: water: formic acid (75:25:0.1). An aliquot of H6 fraction was reduced with a 0.5M TCEP solution, then desalted with C18 ZipTip^®^.

A small amount (2–6 µL) was introduced into an Orbitrap Velos mass spectrometer, equipped with ETD module (Thermo Fisher Scientific, Bremen, Germany) using a TriVersaNanoMate^®^ (Advion Biosciences, Ithaca, NY, USA) in positive ion mode. The spray voltage was set at 1.2–1.6 kV and the backpressure at 0.3–0.4 psi. A full set of automated positive ion calibrations was performed immediately prior to mass measurement. The transfer capillary temperature was set to 100 °C, the sheath and the auxiliary gasses switched off, and the source transfer parameters optimized using the auto tune feature. The FT automatic gain control (AGC) was set at 1 × 10^6^ for MS experiments. The spectra were acquired in the FTMS in full profile mode with 10 micro-scans 1 min, with averaging on and set to max, and a resolution of 30,000 at *m*/*z* 400. For MS/MS experiments, the FT automatic gain control (AGC) was set at 2 × 10^5^. Ions corresponding to the isotopic distribution of a single charge state were selected with the largest possible window, to avoid overlap with neighboring species but minimize signal loss. HCD (High Collision Dissociation) was performed at 30 eV and spectra were acquired in the FTMS in full profile mode at a resolution of 30,000 at *m*/*z* 400, with 10 microscans and with averaging on and set to the maximum value. The spectra were averaged using Qualbrowser in ThermoXcalibur 2.1 and deconvoluted using Xtract to produce zero charge mass spectra, and de novo sequencing was realized manually.

### 3.4. Cell Culture

African green monkey kidney cells (Vero, ATCC CCL 85) were grown as monolayers in Dulbecco’s modified Eagle’s medium (DMEM) supplemented with 10% fetal bovine serum (FBS) (Gibco, Cergy-Pontoise, France) and incubated at 37 °C with 5% CO_2_ in tissue culture flask, until total confluency.

### 3.5. Vero Cell Viability Assay

The effect of the venom peptide on cell viability was determined by the MTT (3-[4,5-dimethylthiazol-2-yl]-2,5 diphenyl tetrazolium bromide) assay. Vero cells were grown at a cell density of 2 × 10^5^ cells/well, in 96 well plates, and cultured at 37 °C for 24 h. The peptide was added to the wells at different concentrations (from 100 µg/mL to 5 µg/mL). After 72 h incubation, the medium was removed and 100 µL of MTT solution (5 mg/mL in PBS was added to each well, then re-incubated for 4 h at 37 °C. The supernatants were aspired out and 100 µL DMSO was added to solubilize the formazan crystals in each well. The absorbance was monitored in a reader plate (MULTISKAN, Labsystems, Vantaa, Finland) at 560 nm.

### 3.6. Virus Titration

The viral titer was assessed using a plaque reduction assay. Vero cell monolayers, seeded in 24 well plates, were infected with 10-fold dilutions of NDV suspension. After virus adsorption for 1h at 37 °C, the inoculum was removed. The cells were washed three times with PBS and overlaid with a medium containing 1.2% methylcellulose. After 72 h incubation at 37 °C, cells were fixed and stained with 1% crystal violet containing 20% methanol. The plaque viral titers were counted and expressed as plaque forming units (PFU), calculated according to the plaque numbers and dilution ratio [24].

### 3.7. Viral Inactivation Assay

Screening of venom fraction on NDV inactivation was carried out by plaque reduction assay. Vero cell monolayers were cultivated in 24-well plates (10^5^ cells/well) and incubated at 37 °C in 5% CO_2_. Each fraction was pre-incubated, at 10 µg/mL, with NDV (MOI = 0.001) for 2 h at 37 °C, then added to the Vero cell layers. After 2 h of incubation, cells were washed with PBS and overlaid with a medium containing 1.2% methylcellulose. After 72 h incubation at 37 °C, cells were fixed and stained with 1% crystal violet containing 20% methanol to count viral plaques. All antiviral assays were repeated three times. The concentration reducing 50% (EC_50_) of the plaque forming units was calculated by comparing treated and untreated cells (100% infectivity, as positive control), using GraphPad Prism software Version 4.0.

The most active peptide was tested at different concentrations (10 µg/mL; 5 µg/mL; 2.5 µg/mL; and 1.25 µg/mL). This peptide was then tested at the highest concentration (10 µg/mL), as described by Zhengyang et al. [17], to study its effect on the different steps of the multiplication cycle of NDV.

#### 3.7.1. Cell Protection Assay (Pretreatment Assay)

Vero cells were incubated in presence of the venom peptide for 2 h at 37 °C in 5% CO_2_. Subsequently, the cells were washed with PBS and infected with NDV at 0.001 MOI. After 2 h incubation, cells were washed with PBS then replenished with a cover layer as described above.

#### 3.7.2. During Infection Assay

Vero cells were infected with the NDV (0.001 MOI) in the presence of the venom peptide (10 µg/mL), then incubated at 37 °C. After 2 h incubation, the cell layers were washed three times with PBS, and then replenished with a cover layer.

#### 3.7.3. Post Treatment Assay

Vero cells were infected with NDV at MOI 0.001 for 2 h at 37 °C. Afterwards, cells were washed three times with PBS to remove unbound virus and then a cover layer containing the venom peptide was added.

### 3.8. Hemolysis Assays

The hemolytic activity of the venom peptide was assessed using chicken erythrocytes, according to the method of Maston [25]. Blood was centrifuged for 5 min at 250× *g* and washed three times with PBS, then resuspended to make a solution of 1% erythrocytes for the hemolytic assay. Serial dilutions from 100 µg/mL to 5 µg/mL of the scorpion venom peptide were mixed with the erythrocytes suspension and incubated for 45 min at 37 °C, then centrifuged at 250× *g* for 5 min at 4 °C. The absorbance of the supernatants was measured at 545 nm. The negative control consisted of treatment with sterile PBS, while the positive control consisted of treatment with 0.1% Triton X-100.

### 3.9. In Ovo Toxicity Assay

The venom fraction (BotG50) as well as the active peptide, were evaluated at different concentrations for their effect on specific pathogen-free (SPF) embryos (ValoBio Media GmbH, Osterholz-Scharmbeck, Germany). The fraction/peptide were diluted in cell culture medium (DMEM) at 10 µg/mL, and then inoculated into 9-day-old embryonated SPF eggs via the allantoic route. Treated eggs were incubated at 37 °C and 80% humidity with a daily candling to examine the embryos’ viability. Dead embryos were immediately stored at 4 °C. The in ovo toxicity assay was repeated three times. After 5 days of incubation, the chicken embryos were photographed and examined.

### 3.10. In Ovo Antiviral Activity

The selected peptide was tested for its effect in reducing the infectivity of NDV in ovo. A mixture of 100 μL of the scorpion peptide (at 10 μg/mL) and 100 μL of NDV at 10^3^ TCID 50/mL (Median Tissue Culture Infectious Dose/mL) was prepared and incubated at 37 °C for 2 h, then inoculated in the allantoic cavity of 9-day-old embryonated SPF eggs. Embryonated eggs inoculated with only the NDV were used as control. Treated SPF eggs were incubated at 37 °C and 80% humidity for 5 days with a daily candling to examine the embryos’ viability. Dead embryos were immediately stored at 4 °C. The allantoic fluids of live and dead embryos were aseptically harvested and stored at −80 °C until used for infectivity titration. To determine the embryo infectious dose (EID_50_) titer, 200 µL of serial dilutions (10^−1^ to 10^−10^) of the harvested allantoic fluid of eggs treated with the scorpion peptide were inoculated to 9-day-old SPF embryonated eggs and incubated for 5 days at 37 °C and 80% humidity with daily candling. At the end of the incubation, the standard hemagglutination assay (HA) was carried out on harvested allantoic fluids to detect the presence of the viral antigens. The EID_50_ was calculated according to the method of Reed and Muench [26].

### 3.11. Hemagglutination Assay

The harvested allantoic fluid was added to the first well of a 96-well plate already containing 25 µL PBS per well. A two-fold dilution was prepared by mixing the suspension and transferring 25 µL to the next well. A volume of 25 µL of 5% (*v*/*v*) chicken blood suspension was added to each well and mixed gently before incubating the plate for 40 min at room temperature. Experiments were performed in triplicate using allantoic fluids from three eggs per treatment.

### 3.12. Sequence Analysis

In order to check the structural relationship between the chlorotoxin-like peptides and the reported scorpion venom AMPs, the sequences of all chlorotoxin-like peptides were first aligned only with antimicrobial scorpion venom peptides containing cysteines using the ClustalW program (https://www.genome.jp/tools-bin/clustalw (accessed on 5 December 2022)) according to the default parameters (fast/rough alignments). The Clustalw starts with a pairwise sequence alignment, and then builds the tree of evolutionary relations between the sequences. A complementary phylogenetic analysis was then carried out by introducing antimicrobial peptide and antiviral sequences from scorpion venom with or without cysteines. The phylogenetic tree was generated by the Mega software package (version 3.1), according to the JTT (Jones–Taylor–Thornton) model. The algorithm of this template allows for sequence comparison based on approximate peptides, and defined sequences are clustered at the 85% level of identity. The closest sequence pairs are aligned, and the observed acid exchanges are taken into account in a matrix. All peptide sequences were taken from published reviews [27,28].

### 3.13. Protein–Protein Docking (In Silico Analyses)

In this study, docking analysis was established by a docking algorithm of HDOCK software (http://hdock.phys.hust.edu.cn/ (accessed on 21 January 2023)) to predict structures of the complex scorpion toxin-HN of NDV. The crystal structure of the bind protein HN-NDV (PDB accession 3T1E) was taken from the Protein Data Bank (https://www.rcsb.org/structure/3t1e (accessed on 21 January 2023)). The 3D structure model of AaCtx was obtained as reported in [20]. Water molecules as well as other hetero-atoms were removed from the crystal structure of the HN-NDV structure before proceeding with the docking stage. The HDOCK server, for integrated protein–protein docking, automatically predicts their interaction through a hybrid algorithm of template-based and template-free docking. Both the cleaned bind protein and the ligand files were loaded on the web server and docked according to the default parameters. The top ten solutions, ranked according to the docking scores, were retained to select the best one, according to its binding site in HN-NDV, the binding affinity (ΔG) and the dissociation constants (Kd), calculated with the PRODIGY program [29,30]. The visualization of complexes as well as the electrostatic charge, were established with the PyMOL software (version 2.5).

### 3.14. Statistical Analysis

Statistical analysis was carried out with one-way ANOVA followed by Bonferroni test, using Graph-Pad Prism software (version 4.0). Numerical results were one representative of experiments repeated at least three times expressed as mean ± standard error of the mean (SEM). Significance was set at the 95% level. Statistical significance levels were defined as * *p* < 0.01, ** *p* < 0.001 and *** *p* < 0.0001. The EC_50_ and CC_50_ values were calculated with regression analysis using the software GraphPad Prism Version 4.0 (GraphPad Software, San Diego, CA, USA) by fitting a variable slope-sigmoid dose–effect curve. The selectivity index (SI) was calculated by dividing the CC_50_ by the EC_50_ value.

## 4. Discussion

The development of new AMPs can expand the existing databases of antimicrobials and give hope for a possible solution for animal and human health, in the face of emerging pandemics and antimicrobial resistance. The glycoprotein hemagglutinin-neuraminidase (HN) on the surface of the NDV is responsible for cell attachment, the promotion of fusion and the release of progeny virions. This multifunctional nature of HN makes it an attractive target for the development of inhibitors of NDV.

Interestingly, many amino acid residues of the HN-NDV binding site are conserved with that of human parainfluenza HN, suggesting that it could be used as a model for the structure-based design of potential inhibitors as treatments for different diseases [22].

Many research studies have shown that scorpion venom compounds possess antiviral activities in vitro, and are considered as promising tools for developing effective antiviral drugs [31,32]. Hp1090 was the first reported natural scorpion venom antiviral peptide. It is an NDB peptide discovered through cDNA technology from the gland of *Heterometrus petersii* scorpion. This peptide inhibits Hepatitis C virus infection in vitro with an IC_50_ value of 7.62 µg/mL [33]. Afterward, several other scorpion venom antiviral peptides were reported. However, no study has been reported for scorpion venom peptides with anti-NDV activity. In Tunisia, the *Buthus occitanus tunetatus* (*Bot*) scorpion is the most represented species and its venom is considered to be the most toxic after that of *Androctonus australis* species. In this study, we screened the venom of *Bot* scorpion and isolated an active peptide BotCl, with an antiviral effect against NDV. It exhibited a significant dose-dependent infectivity inhibition when incubated with NDV for 2 h at 37 °C, then inoculated to Vero cells with an IC_50_ of 2.5 µg/mL (0.69 µM) (Figure 3B).

In particular, we have demonstrated that BotCl acted directly on NDV to inhibit its viral infectivity (Figure 5). BotCl acts by disrupting the structure of the virus particles, resulting in their inactivation and inability to infect host cells. This mechanism of action is effective against NDV, as it directly targets the virus particles outside of the host cells, preventing their entry into the cells and their multiplication.

Interestingly, the MTT assay showed that BotCl has no cytotoxic effect on Vero cells up to a concentration of 100 µg/mL (Figure 3A). Furthermore, at this concentration, no hemolytic activity was observed when added to chicken blood (Appendix A), contrary to the majority of scorpion AMPs, whose therapeutic value is limited by their hemolytic activity [31]. These interesting results prompted us to evaluate the antiviral activity of BotCl in ovo. We found that BotCl exerted a significant inhibition on NDV growth by decreasing its viral titer in SPF chicken embryos by 73% (Figure 7). Thus, all these activities demonstrate that BotCl has an antiviral activity.

On the other hand, according to the purification scheme, we noted that the fraction containing BotCl was retained by the Resource S cation-exchange chromatography column, indicating that it carried a positive charge. Furthermore, BotCl was also eluted late in the gradient of acetonitrile from the RP-HPLC column, showing that this peptide is hydrophobic. These characteristics are similar to all the AMPs, especially those identified from scorpion venoms.

Interestingly, LC/MS analysis revealed the presence of four disulfide bonds, thus classifying BotCl in the family of DB-AMPs.

Interestingly, the molecular mass and the four disulphide bridges highlighted by mass spectrometry, as well as the N-terminal sequence of BotCl, showed that it belongs to the Chlorotoxin-like peptides family (Figure 2).

Disulfide bonds often occur within well-defined structural motifs (such as the CSα/β or ICK motif), and typically have a role in stabilizing peptides by reducing their conformational flexibility. The disulfide bond connectivity in chlorotoxin was reported to be more like a CSα/β motif than an ICK motif [34].

Chlorotxin-like peptides have been reported to be an interesting template for the development of diagnostic and therapeutic agents for cancer, but as far as we know no antimicrobial effect was reported for any of these peptides. Thus, BotCl is the first chlorotoxin-like peptide shown to exhibit antiviral activity.

Since we have already isolated AaCtx, another chlorotoxin-like peptide from scorpion Aa venom, we tested it at a concentration of 10 µg/mL and found that it had a three-fold lower effect than BotCl at the same concentration. This result suggests that the other chlorotoxin-like peptides may have an anti-NVD effect, with different affinities.

Given the lack of anti-NDV scorpion peptides, the activity of BotCl could be compared to the few identified disulfide-bridged antiviral peptides such as Smp76, isolated from *Scorpiomaurus palmatus*, which inhibits the ability of Hepatitis C virus (HCV) to infect the host cells with an IC_50_ of 0.01 µg/mL [28]. EV37 is a scorpine-like peptide of 78 amino acid residues, including the CSα/β motif. It inhibited DENV-2 infection and suppressed HCV and ZIKV infections with IC_50_ of 10 µM [35]. BmKDfsin3 is composed of 38 amino acid residues, with 3 disulfide bonds. It inhibits HCV replication and affects the attachment and post-entry stages of the viral infection cycle with an IC_50_ of 3.35 µM [27]. The BmKDfsin4 inhibits the activity of HBV in HepG2.2.15 cells with IC_50_ values of 1.26–3.95 µM against the production of extracellular HBsAg and HBeAg. Thus, with its IC_50_ of 0.69 µM, BotCl could be considered as a potent antiviral scorpion toxin, compared to these peptides.

However, there remains the question regarding the functionality of BotCl in *Bot* scorpion venom, knowing that scorpions are not subject to NDV infections. Scorpion venomous gland contains a wide range of biological active molecules, including AMPs [3,4], that are believed to be an integral component of an innate immune system that serves to protect the scorpion and its gland against a variety of pathogens [36,37] and facilitate the action of other neurotoxins [11]. Consequently, and given the conservation of haemagglutinin neuraminidase glycoproteins on the surface of different viruses, BotCl could probably block other viral infections.

As both BotCl and AaCtx from the chlorotoxin-like peptides family showed an antiviral effect, and assuming that structural homology leads, generally, to the exhibition of homologous functional properties, we can presume that the other chlorotoxin-like peptides could also have antiviral effects. Thus, we studied the potential structural relationship of the Chlorotoxin-like peptides with the reported scorpion venom DB-AMPs. The sequence alignment showed that the primary structures of Chlorotoxin-like peptides could be definitely aligned with those of the scorpion venom DB-AMPs. Particularly, we found that five of the eight cysteine residues of the chlorotoxin-like peptides were conserved in all the sequences. Another cysteine residue (Cys5 of ClTx)was also conserved, except with HgebKTx. Thus, according to the sequence alignment reported in Figure 8A, it is obvious that the chlorotoxin-like peptides are structurally related to the reported scorpion venom DB-AMPs. The reinforcement of the arguments in favor of the structural relationship between Chlorotoxin-like peptides and AMPs was taken advantage of by a phylogenetic tree (Figure 8B). Indeed, it is obvious that the evolutionary relationship between the sequences of chlorotoxin-like peptides and the sequences of scorpion venom DB- or NDB-AMPs shown in Figure 8B is one of the effective ways to prove the existence of a new family of AMPs represented by the ClTx.

Since AaCtx has shown an inhibiting effect on NDV infectivity, and in order to identify and characterize the putative mode of interaction between the chlorotoxin-like peptide and the NDV, a blind docking was carried out with the crystal structure of the catalytically active head region (residues 124 to 569) of HN from the Kansas strain of NDV [7]. HN has a uniquely large cavity around the O4 position of its substrate, which is lined with residues that are largely conserved across all HNs and play a functional role of fusion upon binding receptors [38,39].

The most reliable solution of the docking showed that AaCtx binds the cavity of the HN-NDV binding site, mainly with the loop connecting the α-helix to the β-sheet (Figure 9A). The most important interaction was that of G21 of AaCtx with R416 (Figure 9B). Interestingly, this later belonged to the tri-arginyl cluster (R174, R416 and R498), identified as a crucial triad for the activity of HN-NDV, and their mutations severely reduced HN activity [22]. Interestingly, R498 is also implied in an important interaction with A13. Other interactions are important in the complex, such as that of R23 and R24 with E401/S237 and E258 of HN-NDV, respectively. In fact, the E258 and Y262 lie on a helix that is stabilized by the calcium ion, and it is interesting to note that the removal of Ca^2+^ completely abolishes enzyme activity [22].

Interestingly, the AaCtx interaction with HN-NDV could be compared to that of Neu5Ac2en, a HN inhibitor [22]. This inhibitor binds the tri-arginyl residues, as well as I175, S237, E401, E547, E258 and Y526 of the HN active site [22,40]. These interactions were found in AaCtx, mainly by its R23 (with I175, S237 and E401), S22 (with R416, E401 and Y526 Y526) and R24 (with E258) residues. Thus, the interaction model of AaCtx with NDV was in accordance with its antiviral effect.

## 5. Conclusions

In the light of these results, BotCl is obviously a potent anti-NDV peptide that may be considered a promising tool to study and control the pathogenicity of the Newcastle disease virus, as well as other, similar viruses such as avian or human influenza virus. Although the complete sequence of the BotCl peptide has not been determined, its N-terminal sequence is sufficient to assert that this peptide belongs to the chlorotoxin-like peptides family. Interestingly, the antiviral activity of BotCl, as well as that of AaCtx, in addition to the sequence alignment of chlorotoxin-like peptides and the model of interaction of AaCtx with HN-NDV, converge towards the high probability of the chlorotoxin-like peptides family belonging to that of AMPs. Thus, this study highlights the emergence of a new family of AMPs which provide promising prospects for investigating this peptide’s family for a new, interesting activity for the development of a new generation of antimicrobial products.

As prospective, more refined works with cloned and recombinant expressed BotCl in bacterial models, conducive to the synthesis of disulfide bridges, will make it possible to obtain it in large quantities, this will allow its study in adult animals exposed to nature conditions and wild type virus. Furthermore, recombinant mutated analogs of BotCl could allow for a better study of the structure–activity relationship of chlorotoxin-like peptides and the mechanism of their interaction with NDV.

## Figures and Tables

**Figure 1 molecules-28-04355-f001:**
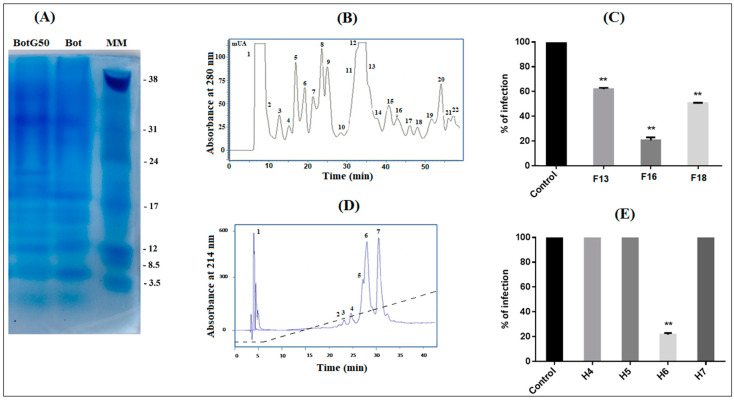
Screening of *Bot* scorpion venom and purification of the most active molecule. (**A**) The SDS-PAGE of the crude *Buthus occitanus tunetanus (Bot*) scorpion venom (50 µg) and the BotG50 fraction (50 µg), obtained from its G50 filtration. (**B**) The BotG50 fraction was injected to a cation exchange FPLC Resource S column equilibrated with 0.05 M ammonium acetate, PH = 6.7. The molecules were eluted at a constant flow rate of 1 mL/min, using a linear gradient of 58 min, from 0.05 to 0.5 M ammonium acetate. Peaks were detected at a wavelength of 280 nm. (**C**) Fractions obtained by FPLC were tested for their antiviral effect. Only active fractions were represented. (**D**) The most active fraction F16 was fractionated by RP-HPLC, as described in the experimental section. (**E**) Only H6 showed an anti-NDV effect. Error bars represent the standard error of the mean of three independent experiments. ** *p* < 0.001.

**Figure 2 molecules-28-04355-f002:**
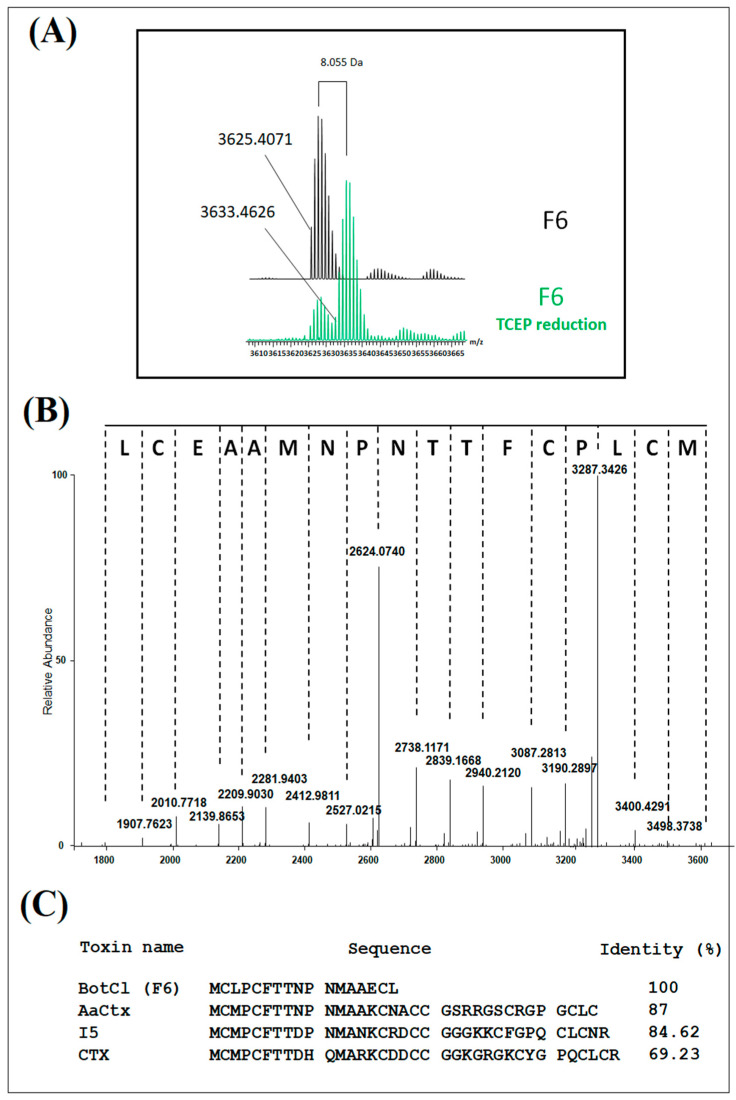
Structural characterization of BotCl. (**A**) Mass spectrometry analysis of BotCl before and after reduction with TCEP deconvolution. (**B**) HCD MS/MS spectrum of BotCl. (**C**) N-terminal sequence of BotCl and comparison with those of other chlorotoxin-like peptides.

**Figure 3 molecules-28-04355-f003:**
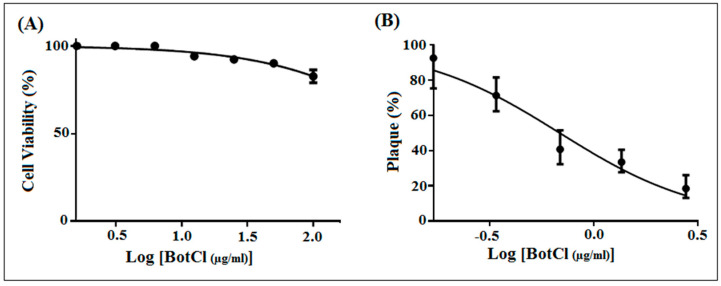
In vitro dose–response effect of BotCl. (**A**) Effect of different concentrations (ranging from 5 to 100 μg/mL) of BotCl on Vero cells’ viability, measured by MTT assay. (**B**) Effect of different concentrations (10 µg/mL; 5 µg/mL; 2.5 µg/mL; and 1.25 µg/mL) of BotCl on NDV inactivation. NDV was incubated with BotCl then applied on Vero cells. The inhibitory effect was determined using a plaque reduction assay. Error bars represent standard error of the mean of three independent experiments.

**Figure 4 molecules-28-04355-f004:**
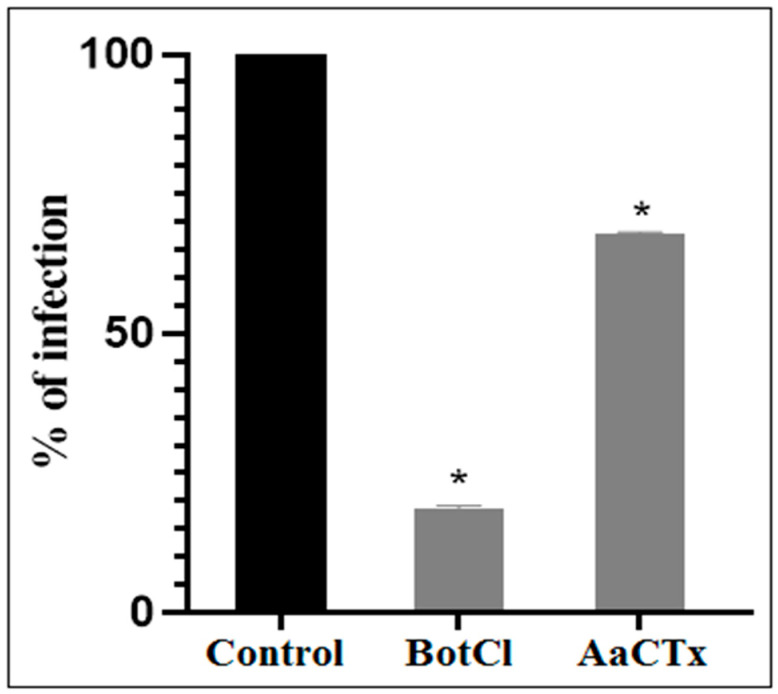
Effects of BotCl and AaCTX on NDV infectivity: a mixture of BotCl or AaCTX, at 10 µg/mL, and NDV was incubated for 2 h at 37 °C and then inoculated to Vero cells. The inhibitory effects of BotCl and AaCtx were determined with a plaque reduction assay. Error bars represent standard error of the mean of three independent experiments. * *p* < 0.01.

**Figure 5 molecules-28-04355-f005:**
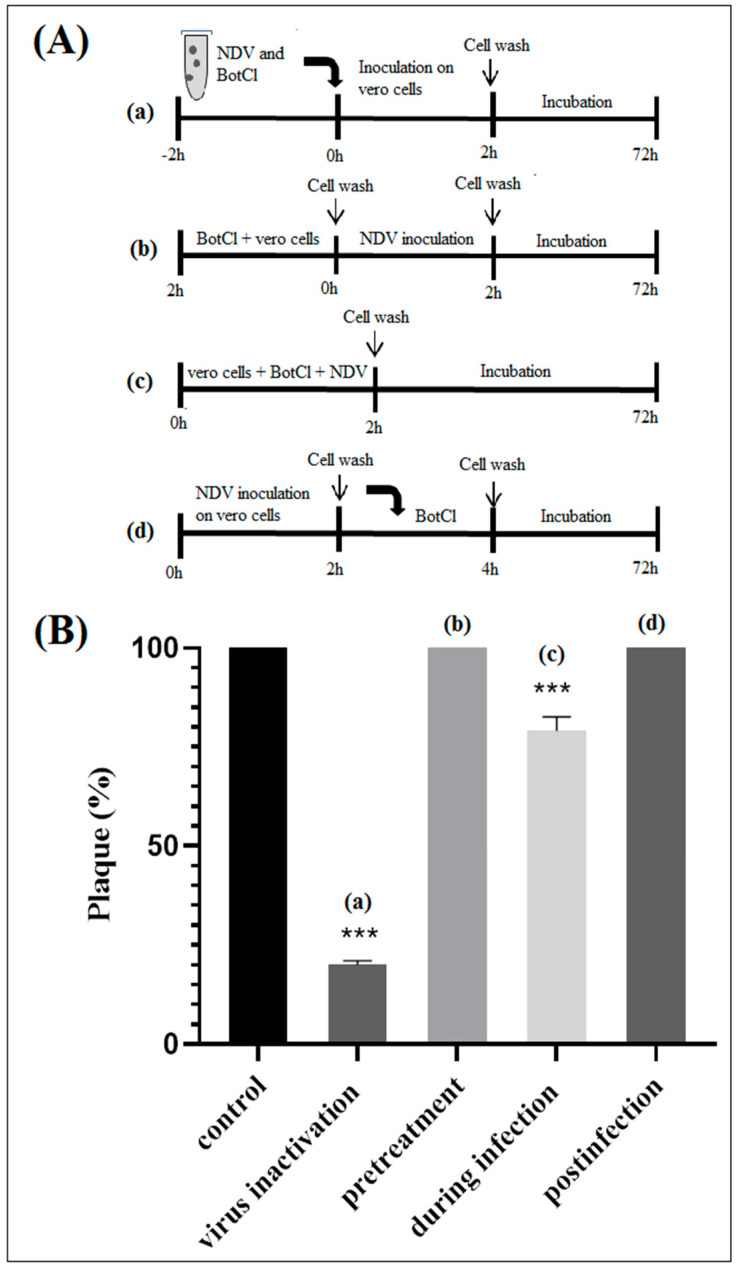
In vitro effect of BotCl against NDV. (**A**) Schematic overview of time of addition experiments. BotCl was used in four different treatment modes: (a) inactivation assay, (b) pretreatment, (c) during infection and (d) postinfection assays with NDV on Vero cells (see experimental section). (**B**) Antiviral activity of BotCl peptide in time of addition assay. The cells or viruses were treated with BotCl peptide at a final concentration of 10 µg/mL. The inhibitory effects of BotCl in each treatment mode were determined using a plaque reduction assay. Error bars represent standard error of the mean of three independent experiments. *** *p* < 0.0001.

**Figure 6 molecules-28-04355-f006:**
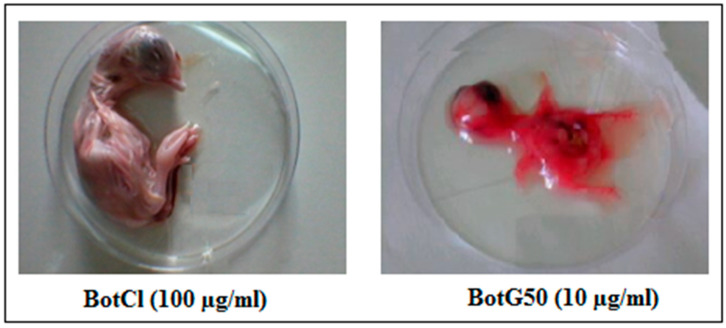
Effect of BotCl and BotG50 fractions on embryonic chicken.

**Figure 7 molecules-28-04355-f007:**
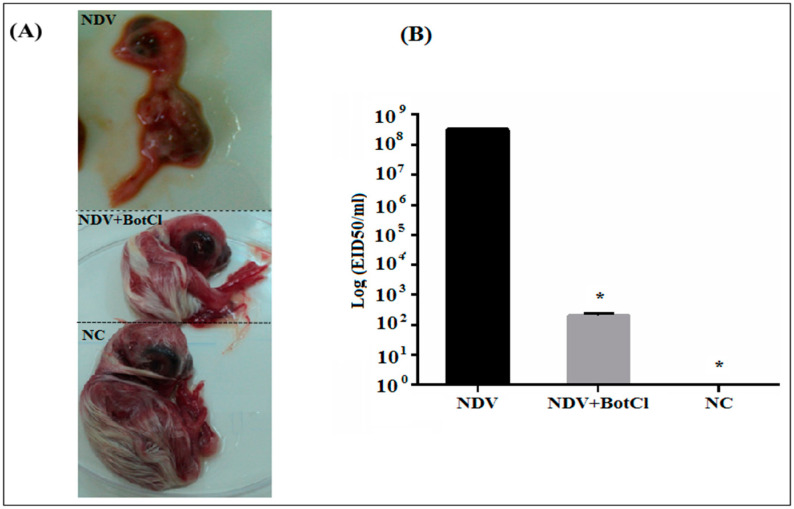
In ovo effect of BotCl on NDV infectivity. Eggs were inoculated with a mixture of BotCl (10 µg/mL) and NDV after 2 h incubation at 37 °C. (**A**) Effect on chicken embryo growth. (**B**) Titration of NDV infectivity in allantoic fluid of embryonic chicken. Negative control (NC) corresponds to untreated eggs and positive control corresponds to embryonic chicken inoculated with NDV only. Error bars represent standard error of the mean of three independent experiments. * *p* < 0.01.

**Figure 8 molecules-28-04355-f008:**
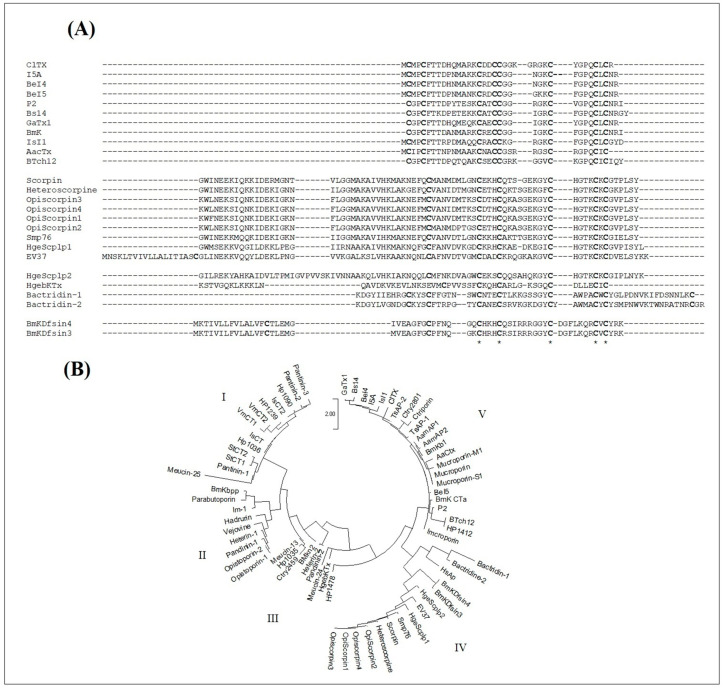
Structural relationship study of chlorotoxin-like peptides with reported scorpion venom antimicrobial peptides. (**A**) Sequence alignment with scorpion venom AMPs containing cysteines. * correspond to identical conserved residues in all the sequences (**B**) Phylogenetic analysis of chlorotoxin-like peptides versus reported scorpion venom AMPS containing or not containing cysteines, generated by the Mega 3.1 software package according to the JTT model. The scale bar represents a genetic distance of 2.

**Figure 9 molecules-28-04355-f009:**
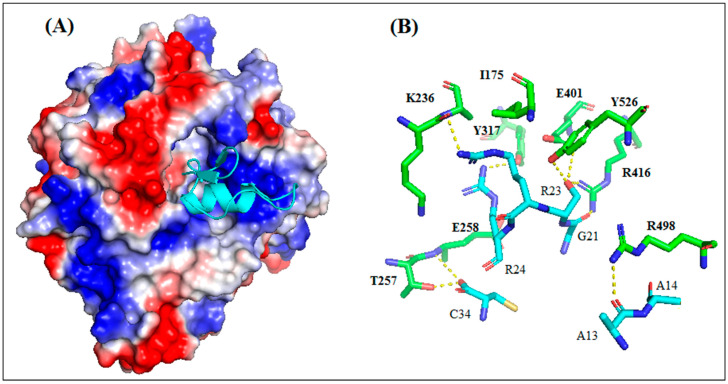
Interaction mode of AaCtx with HN of NDV. (**A**) The peptide interacts with the binding site of HN NDV with multiple points interaction. (**B**) Key interactions of AaCtx residues with HN NDV.

## Data Availability

The authors confirm that the data supporting the findings of this study are available within the article.

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
