# Peer review of "BotCl, the First Chlorotoxin-like Peptide Inhibiting Newcastle Disease Virus: The Emergence of a New Scorpion Venom AMPs Family"

_molecules, 2023, doi:10.3390/molecules28114355_

Round 1
Reviewer 1 Report
In this article, the authors describe the anti-NDV activity of an isolated peptide from scorpion venom. They isolate various fractions of the venom using MS and evaluate them in a Vero cell NDV model, followed by time-of-addition experiments to shine a light on the mode of action of the peptide. They partially sequence the peptide to find similarity with other scorpion venoms, which allows classification as chlorotoxin-like peptide, and perform a phylogenetic analysis on presumed close related peptides.
This is an interesting study, especially since this is the first described chlorotoxin-like peptide with antiviral activity. However, the manuscript needs major revision before it is suited for publication.
Major remarks:
- Controls are missing in the antiviral experiments: the quality of every employed assay should be supported by including NDV inhibitors with described in vitro or in ovo activity. Also, can another chlorotoxin-like peptide be included as a (negative) control?
- The authors clearly state that the fraction of BotCl is too small to sequence. However, what is needed to be able to obtain a full sequence? This is highly necessary because now the 3.7 section is very speculative. If full sequencing is not possible in this manuscript the authors should better expand on future studies needed.
Minor remarks:
- Please thoroughly proofread the manuscript to improve the English.
- Section 3.4 is written somewhat unclear. 315-316 talks about 3 different setups, while 317-323 talk about 4 results. This is also reflected in Fig. 4, where 3 setups are shown in the scheme, while 4 result graphs are shown. It takes too much effort from the reader to get through this part and to understand what is added when. The authors can for example adapt the scheme by including icons where virus/BotCl/cells are added.
- Section 3.4 B: based on the methodology section, in the pretreatment setup the BotCl is washed away before infection. Can the authors also include a condition where it is not washed away? If it has virucidal activity this should also be reflected in decreased infection. With virucidal effect the virus should be compromised to enter the cells if the compound is already present.
- Figure captions: please include the number of technical replicates as well so that is clear how many biological and technical replicates are included. Why do the authors show SEM instead of SD?
- 3.7: add some introduction to this section since it is not clear until the discussion why you did not include BotCl in this analysis.
- Line 434: please include as supplementary information.
- Line 453: did the authors check potential anti-NDV activity of these compounds?
Author Response
Reviewer 1
In this article, the authors describe the anti-NDV activity of an isolated peptide from scorpion venom. They isolate various fractions of the venom using MS and evaluate them in a Vero cell NDV model, followed by time-of-addition experiments to shine a light on the mode of action of the peptide. They partially sequence the peptide to find similarity with other scorpion venoms, which allows classification as chlorotoxin-like peptide, and perform a phylogenetic analysis on presumed close related peptides.
This is an interesting study, especially since this is the first described chlorotoxin-like peptide with antiviral activity. However, the manuscript needs major revision before it is suited for publication.
The authors thank the reviewer for their generally positive assessment of our
manuscript and for pointing out some oversights, for which we apologize. We have provided point-by-point responses to the issues raised in the following and revised text and figures, accordingly.
Major remarks:
- Controls are missing in the antiviral experiments: the quality of every employed assay should be supported by including NDV inhibitors with described in vitro or in ovo activity. Also, can another chlorotoxin-like peptide be included as a (negative) control?
We agree with the reviewer. Indeed we missed a positive control in our experiments. In fact, there is no active molecule on NDV described until now, that's why we couldn't have a control with effective drug against NDV virus. The native AaCtx, a chlorotoxin-like peptide from Aa scorpion, was tested and the result showed that it is active, even though it showed lesser effect than BotCl. This is now added in the revised manuscript
- The authors clearly state that the fraction of BotCl is too small to sequence. However, what is needed to be able to obtain a full sequence? This is highly necessary because now the 3.7 section is very speculative. If full sequencing is not possible in this manuscript the authors should better expand on future studies needed.
We completely agree with the reviewer. We should try to complete the sequence of BotCl by designing a primer of nucleotides and try to clone the gene of BotCl from the venom gland. However, we can have scorpions only in summer and this will take months of study. It can be the subject of future study. This is now added in the manuscript as perspectives. On the other hand, we think that the 3.7 section is no longer speculative since we showed that AaCtx has an inhibiting effect on NDV infectivity.
Minor remarks:
- Please thoroughly proofread the manuscript to improve the English.
We apologize for the English errors. As suggested by the reviewer, the manuscript was thoroughly proofread and improved and we hope that it is more clear and easy to follow by the reader.
- Section 3.4 is written somewhat unclear. 315-316 talks about 3 different setups, while 317-323 talk about 4 results. This is also reflected in Fig. 4, where 3 setups are shown in the scheme, while 4 result graphs are shown. It takes too much effort from the reader to get through this part and to understand what is added when. The authors can for example adapt the scheme by including icons where virus/BotCl/cells are added.
After taking your comments into consideration, we have made modifications to Section 3.4. It has been reformulated to provide a clear description of the results. Furthermore, a schematic representation of the virucidal assay has been added in the figure 5 (Figure 4 in the precedent version), and the order of the bars in the histogram has been modified to align with the changes made to the materials and methods described in the text.
- Section 3.4 B: based on the methodology section, in the pretreatment setup the BotCl is washed away before infection. Can the authors also include a condition where it is not washed away? If it has virucidal activity this should also be reflected in decreased infection. With virucidal effect the virus should be compromised to enter the cells if the compound is already present.
Authors thank the reviewer for this remark. As the reviewer mentioned in the pretreatment setup, the BotCl is washed before infection to eliminate any venom peptide that did not enter or bind to the cells. Therefore, there is no free BotCl when the virus is added to cells, since there is no interaction between BoCl and Vero cells. That’s why we didn’t notice any decrease of the infectivity. In all treatment protocols, cells are washed after each incubation in order to eliminate any non bonded compound to cells or virus to gain insights into the mode of action of BotCl.
- Figure captions: please include the number of technical replicates as well so that is clear how many biological and technical replicates are included. Why do the authors show SEM instead of SD?
We have chosen to use SEM with a replication of 3 to allow easy comparison of variability between different tests. The SEM is more suitable for comparing results across different conditions and is not influenced by the sample size.
- 3.7: add some introduction to this section since it is not clear until the discussion why you did not include BotCl in this analysis.
We thank the reviewer for this remark. We added an introduction in this section 3.7.1
- Line 434: please include as supplementary information.
The figure is included as Figure S1 in supplementary data
- Line 453: did the authors check potential anti-NDV activity of these compounds?
We also think that checking the potential anti-NDV activity of these compounds will be very interesting. However, we couldn't have these molecules, they’re not even available on the toxins market

Reviewer 2 Report
Reference: “BotCl, the first Chlorotoxin-like peptide inhibiting Newcastle disease virus: emerging a new scorpion venom AMPs family” Abir Jlassi et al., 2023, submitted to Molecules, 2023.
General comments: In the submitted text for publication in the journal MDPI/MOLECULES, the authors are reporting their experimental data describing the identification of a peptide found in the venom of the scorpion Buthus occitanus tunetanus with inhibitory activity on the development of the Newcastle disease virus. Newcastle disease is a viral disease that plagues birds in general and around the world, and although it seems not to affect humans, it has brought destruction to agricultural herds and great economic losses. Advances in this area are very welcome and deserve to be evaluated seriously and professionally. Throughout the text, the authors identify a biologically active peptide against viral particles with an IC50 of 0.69 µM, and a low cytotoxicity on cultured Vero cells (CC50 > 55 µM). Furthermore, tests on uninfected embryos showed that treatments with this peptide had a protective effect and reduced by 73%, the virus titer in allantoic fluid. The peptide was named “BotCl” due to its identity with members of the Chlorotoxin Family found in scorpion venoms. The authors also argue that “BotCl” is the first peptide found in scorpion venoms capable of blocking the binding of viral particles to cell receptors. After careful reading of the text, it is my opinion that it is a good text, with the potential to be published in Molecules. However, here are some suggestions and questions that I would like the authors to consider in a revised version to improve the quality and attractiveness of the text.
Specific Comments:
1- In Key Words, lines 32 and 33 …. Newcastle disease virus; antiviral activity; in vitro; in ovo; scorpion venom peptides; Buthus occitaus tunetanus . I suggest a clearer line of word organization. Something like: …. Buthus occitaus tunetanus venom, Chlorotoxin-like peptide “BotCl” , antiviral activity, Newcastle disease virus.
2- Line 38 … worldwide [1] [2]. Please change [1] [2], by [1,2].
3- In my humble opinion, the introduction is very well written, clear, succinct, and discussing what matters to understand the topic. I would add, to increase the attention of readers, a figure 1A composed of a photograph of a scorpion of the Genus/species studied, a Figure 1B with a Tricine-SDS-PAGE electrophoresis protein profile of crude venom, and a figure 1C with the structure of the viral particle, with its main proteins, which may be useful later in the discussion of this same text, in addition to showing the molecules indicated in this introduction (lines 41 to 48).
4- In the Materials and Methods chapter, the authors must mention the cities and countries where the reagents used throughout the work were purchased. This is the rigid rule of the scientific text, and it was not followed. Example... Between lines 71 to 79 …. African green monkey kidney cells (Vero, ATCC CCL 85) were purchased from the American Type Culture Collection (ATCC), please indicate city and Country. Live virus vaccine ND (LaSota strain) was used in all the assays (supplier address ?)…. Cell culture supplements and reagents were purchased from GIBCO (Cergy-Pontoise, France, here is right)… UltraPure™ Agarose was from Invitrogen ??. Plese see all the chapter of M/M and complete for the first time of citation! Sephadex G-50….and so on….
5- Lines 81 to 85 …Purification Chapter… authors must complete the details of chromatography performed to others reproduce their results. Elution speed of fractions, volumes of collected samples, chromatography temperature, size of chromatography column are important details!
6- In the line 85 and also through all of the text, please change “molecular weight” by “molecular mass” since a molecule has mass instead weight. We perform mass spectrometry instead weight spectrometry!!! This term although used is not correct!!! Weight is for rice, bread, meat, …
7- Still between lines 80 to 97 and through out of the text, it is a good idea to indicate the names of the manufacturers of the equipments used along the procedures!
8- In the lines 126 and 127 change … Methylthiazolyldiphenyl-tetrazolium bromide (MTT) by MTT (3-[4,5-dimethylthiazol-2-yl]-2,5 diphenyl tetrazolium bromide) .
9- In the lines 127 to 128 ….The authors wrote …. Vero cells were seeded at a concentration of 2×105 cells/well, … in a revised version of the text better change to … Vero Cells were grown at a cell density of 2×105 cells/well ….
10- In the line 145 the authors wrote ….Vero cell monolayers were seeded in 24 well plates (105 cells/well). Please change to …. cells were grown or cultivated, since cells are not seeds. This term although used is not the more correct.
11- In the line 146 and lines 153 to 155… the authors wrote ….Venom fractions, at 10 µg/ml, pre-incubated with NDV …. The authors must include an explanation of why they used this concentration of toxin/venom fraction? And other concentrations….?
12- In the line 162 …. Vero cells were infected with a mixture of venom peptide and NDV (0.001 MOI). Please in the revised version of manuscript change to … Vero cells were infected with NDV (0.001 MOI) in the presence of venom peptide (10 µg/ml?).
13- In the lines 167 and 168 ….cover layer containing the venom peptide replaced the culture medium. What authors mean with cover layer? Please detail this phrase!
14- In the lines 172 and 173 … The erythrocytes were resuspended in PBS to make a 1% solution for the hemolytic assay . Cells do not form solutions, they form suspensions. Please correct in the revised manuscript!
15- In the lines 226 and 227 … The crystal structure of the réceptor HN-NDV (PDB accession 3T1E). Please change the word Réceptor by receptor if you keep using this word. In my opinion, receptor is not an appropriated word to refer to this molecule, as it is a virus protein and should be treated as a bind protein. True Receptors can generate a cellular biology effect after being activated. Receptor is the protein target by virus in the cells.
16- In summary, in my opinion the authors need to review the methodologies all around the text and detail some points that are incomplete or may suggest a wrong interpretation!
17- In my opinion, in figure 1, the authors should have shown electrophoresis (Tricine-SDS-PAGE, a technique indicated for separations of low molecular mass peptides) with silver staining, for monitoring the purification steps of the studied peptides. This analytical criterion is lacking at work.
18- In the lines 275 and 276 … After reduction with TCEP, the mass of the peptide showed an increase of 8.005 Da (Figure 2A)… The authors have some explanation for this drastic change in peptide molecular mass after reduction. This could be discussed throughout the text!
19- In the lines 276 and 277 … revealing the presence of 4 disulfide bridges in the structure of the peptide. Are these peptides Knottins with inhibitor cystine knot structures? This could be discussed through the text!
20- In the lines 279 to 281 …. Due to its very low concentration in the venom, we couldn't determine the complete amino-acid sequence of this peptide. But with information of a partial peptide reported as above mentioned, line 279, the authors could design a primer of nucleotides and try to clone the gene of targeted peptide from the venom gland! I would like to know the opinion of authors about this possibility?
21- With reference to figure 3, where the authors show the inhibitory activity of the BotCl peptide on the infection of Vero cells caused by the NDV. It would be elegant for them to show an immunofluorescence reaction for virus particles inside the cells. Thus, it would be possible to have a morphological view of the peptide's inhibitory potential and even quantify the inhibition by a second experimental criterion, increasing the competitiveness of the results.
22- In the lines 321 to 323, Figure 4, the authors wrote… These results suggested that BotCl has a direct activity virucidal on NDV and may prevent initiation of its growth. In my opinion, the data are clear and show inhibition of viral infection and duplication of viral particles within cells. However, it is not possible to state that the peptides have virucidal action (which implies that it causes the direct death of viruses). This can occur due to molecules present in the serum, proteases, among others, which can destroy virus particles out of cells. The peptide apparently binds to virus and prevents them from entering cells. But even that would have to be shown by fluorescent probes, or radioactive labeling, for example.
23- About the experiments shown in figure 5, and text between lines 331 to 339, made to show absence of cytotoxicity on bird embryos. It would be interesting to write in the methodology how many times these experiments were repeated. Also, do not stop at the macroscopic analysis of the embryos, but complete with histopathological studies, and finally measure biochemical markers of liver, kidney, heart and blood cytotoxicity at least, if possible, or other possible markers. This would bring more security to the data shown!
24- About the experiments shown in figure 6, and text between lines 342 to 353, made to show protective activity of peptide against NDV infectivity upon bird embryos. It would be interesting to write in the methodology how many times these experiments were repeated. Also, other markers different of macroscopic analysis as pointed in figure 6A. Histopathological analysis of tissues would be useful for complete conclusion. Finally, authors could test greater concentrations of peptide in order to increase the inhibitory effect pointed.
25- In line 346, and figure 6B, please change CN by NC. That is the correct form in English (Negative control).
26- Regarding the data shown in figure 7, they seem clear and well done to me! However, I missed some discussion by the authors about what would be the role of this peptide for scorpions and other animals that produce them? Are scorpions subject to similar virus infections? Or it's an unexpected non-canonical activity? This could be discussed in the text!
27- About figure 8, although experiments were well done and authors have expertise in the area, they know that these results have a high level of speculation, as the authors do not have co-crystallization data, or molecular dynamic studies, or finally mutated isoforms of toxins that could prove this interaction and inhibit anti viral effects triggered by toxin.
28- Given the conservation of glycoproteins type hemagglutinin-neuraminidase (HN) on the surface of different virus, as discussed in the text (lines 408 to 411) could this peptide block other viral infections?
29- In lines 422 and 423 … In this study, we screened the venom of Buthus occitanus tunetanus scorpion and isolated an active peptide BotCl. Any special reason other than epidemiological and geographic to study this particular scorpion?
30- My comments on results shown in figure 8, and text between lines 493 to 512, should not exclude the figure from the text, but only I would like the authors to be more careful in interpreting the data. They should make it clear in the text that these conclusions need to be refined, with the use of techniques such as co-crystallization and site-directed mutations in the peptide.
31- I missed discussions by the authors about more refined work with cloned and recombinant expressed molecules in bacterial models conducive to the synthesis of disulfide bridges, or models with yeast or insect cells (eukaryotes), where mutated forms at important sites of the molecule could also be produced to better study the mechanistic of peptide/NDV interactions.
32- Alternatively, the authors could obtain synthetic chemical analogues to try to increase the yield in obtaining the peptide of study, and thus better refine the experimental data.
33- Finally, the authors tried to study the protective activity of peptide on adult animals exposed to nature conditions and wild type virus? A kind of data that would be very interesting in order to use this peptide as prophylactic or protector against NDV.
Author Response
Reviewer 2
General comments: In the submitted text for publication in the journal MDPI/MOLECULES, the authors are reporting their experimental data describing the identification of a peptide found in the venom of the scorpion Buthus occitanus tunetanus with inhibitory activity on the development of the Newcastle disease virus. Newcastle disease is a viral disease that plagues birds in general and around the world, and although it seems not to affect humans, it has brought destruction to agricultural herds and great economic losses. Advances in this area are very welcome and deserve to be evaluated seriously and professionally. Throughout the text, the authors identify a biologically active peptide against viral particles with an IC50 of 0.69 µM, and a low cytotoxicity on cultured Vero cells (CC50 > 55 µM). Furthermore, tests on uninfected embryos showed that treatments with this peptide had a protective effect and reduced by 73%, the virus titer in allantoic fluid. The peptide was named “BotCl” due to its identity with members of the Chlorotoxin Family found in scorpion venoms. The authors also argue that “BotCl” is the first peptide found in scorpion venoms capable of blocking the binding of viral particles to cell receptors. After careful reading of the text, it is my opinion that it is a good text, with the potential to be published in Molecules. However, here are some suggestions and questions that I would like the authors to consider in a revised version to improve the quality and attractiveness of the text.
The authors thank the reviewer for their generally positive assessment of our manuscript and for pointing out some oversights, for which we apologize. We have provided point-by-point responses to the issues raised in the following, and revised text and figures, accordingly.
Specific Comments:
1- In Key Words, lines 32 and 33 …. Newcastle disease virus; antiviral activity; in vitro; in ovo; scorpion venom peptides; Buthus occitanus tunetanus. I suggest a clearer line of word organization. Something like: …. Buthus occitaus tunetanus venom, Chlorotoxin-like peptide “BotCl” , antiviral activity, Newcastle disease virus.
Agreed and addressed.
2- Line 38 … worldwide [1] [2]. Please change [1] [2], by [1,2].
Agreed and addressed.
3- In my humble opinion, the introduction is very well written, clear, succinct, and discussing what matters to understand the topic. I would add, to increase the attention of readers, a figure 1A composed of a photograph of a scorpion of the Genus/species studied, a Figure 1B with a Tricine-SDS-PAGE electrophoresis protein profile of crude venom, and a figure 1C with the structure of the viral particle, with its main proteins, which may be useful later in the discussion of this same text, in addition to showing the molecules indicated in this introduction (lines 41 to 48).
The reviewer raises valid points, which we aimed to address by adding a graphical abstract where you found the Buthus occitanus tunetanus (Bot) scorpion species and the structure of the viral particle with its main proteins. Furthermore Tricine-SDS-PAGE electrophoresis protein profile of crude venom and the BotG50 fraction was added in the figure 1 as suggested. A paragraph was also added to explain results obtained in the SDS-PAGE figure.
4- In the Materials and Methods chapter, the authors must mention the cities and countries where the reagents used throughout the work were purchased. This is the rigid rule of the scientific text, and it was not followed. Example... Between lines 71 to 79 …. African green monkey kidney cells (Vero, ATCC CCL 85) were purchased from the American Type Culture Collection (ATCC), please indicate city and Country. Live virus vaccine ND (LaSota strain) was used in all the assays (supplier address ?)…. Cell culture supplements and reagents were purchased from GIBCO (Cergy-Pontoise, France, here is right)… UltraPure™ Agarose was from Invitrogen ??. Please see all the chapter of M/M and complete for the first time of citation! Sephadex G-50….and so on….
Agreed and addressed.
5- Lines 81 to 85 …Purification Chapter… authors must complete the details of chromatography performed to others reproduce their results. Elution speed of fractions, volumes of collected samples, chromatography temperature, size of chromatography column are important details!
More details were added in the text
6- In the line 85 and also through all of the text, please change “molecular weight” by “molecular mass” since a molecule has mass instead weight. We perform mass spectrometry instead weight spectrometry!!! This term although used is not correct!!! Weight is for rice, bread, meat, …
The molecular weight was changed by molecular mass (MM)
7- Still between lines 80 to 97 and through out of the text, it is a good idea to indicate the names of the manufacturers of the equipments used along the procedures!
Agreed and addressed.
8- In the lines 126 and 127 change … Methylthiazolyldiphenyl-tetrazolium bromide (MTT) by MTT (3-[4,5-dimethylthiazol-2-yl]-2,5 diphenyl tetrazolium bromide) .
Done
9- In the lines 127 to 128 ….The authors wrote …. Vero cells were seeded at a concentration of 2×105 cells/well, … in a revised version of the text better change to … Vero Cells were grown at a cell density of 2×105 cells/well ….
Done
10- In the line 145 the authors wrote ….Vero cell monolayers were seeded in 24 well plates (105 cells/well). Please change to …. cells were grown or cultivated, since cells are not seeds. This term although used is not the more correct.
Done
11- In the line 146 and lines 153 to 155… the authors wrote ….Venom fractions, at 10 µg/ml, pre-incubated with NDV …. The authors must include an explanation of why they used this concentration of toxin/venom fraction? And other concentrations….?
We were referred to another team that worked on antiviral peptide derived from scorpion venom, and they also chose 10 µg/ml (Zhengyang, Z.et al., Theranostics 2018, 8 (1), 199-211).
12- In the line 162 …. Vero cells were infected with a mixture of venom peptide and NDV (0.001 MOI). Please in the revised version of manuscript change to … Vero cells were infected with NDV (0.001 MOI) in the presence of venom peptide (10 µg/ml?).
Done
13- In the lines 167 and 168 ….cover layer containing the venom peptide replaced the culture medium. What authors mean with cover layer? Please detail this phrase!
The plaque assay is the indicated method used to measure infectious viruses. The assay involves adding viruses to permissive cells and applying a semi-solid overlay that limits the spread of infection to neighboring cells, while cell death leads to the formation of plaques. After an incubation period that allows the virus to attach to cells, the inoculum is removed. The cells are then covered with a medium containing 1.2% methylcellulose, which is composed of nutrients and a substance necessary for cell viability, forming a semi-solid or gel overlay, called cover layer. These details were now mentioned in section 2.6 ( overlaid with a medium containing 1.2% methylcellulose)
14- In the lines 172 and 173 … The erythrocytes were resuspended in PBS to make a 1% solution for the hemolytic assay. Cells do not form solutions, they form suspensions. Please correct in the revised manuscript!
Agreed and adressed
15- In the lines 226 and 227 … The crystal structure of the réceptor HN-NDV (PDB accession 3T1E). Please change the word Réceptor by receptor if you keep using this word. In my opinion, receptor is not an appropriated word to refer to this molecule, as it is a virus protein and should be treated as a bind protein. True Receptors can generate a cellular biology effect after being activated. Receptor is the protein target by virus in the cells.
We agree with the reviewer, this is changed in the text
16- In summary, in my opinion the authors need to review the methodologies all around the text and detail some points that are incomplete or may suggest a wrong interpretation!
We apologize for any misunderstanding. The methodologies all around the text were revised in the in the current version of the manuscript
17- In my opinion, in figure 1, the authors should have shown electrophoresis (Tricine-SDS-PAGE, a technique indicated for separations of low molecular mass peptides) with silver staining, for monitoring the purification steps of the studied peptides. This analytical criterion is lacking at work.
We agreed with the reviewer and added the Tricine-SDS-PAGE of the crude venom and the BotG50 fraction. A paragraph was also added to explain results obtained in the SDS-PAGE figure.
18- In the lines 275 and 276 … After reduction with TCEP, the mass of the peptide showed an increase of 8.005 Da (Figure 2A)… The authors have some explanation for this drastic change in peptide molecular mass after reduction. This could be discussed throughout the text!
TCEP (tris(2-carboxyethyl)phosphine) is a reducing agent often used to break disulfide bonds within and between proteins according to the chemical reaction below
Owing to a mass increase of 2 Da per disulfide reduction, the number of disulfide bonds can be directly determined by a comparison of the molecular weights (MWs) of the reduced vs. non-reduced intact proteins (Zhao et al. 2013). This is now more explained in the result section.
19- In the lines 276 and 277 … revealing the presence of 4 disulfide bridges in the structure of the peptide. Are these peptides Knottins with inhibitor cystine knot structures? This could be discussed through the text!
Disulfide bonds often occur within well-defined structural motifs (such as the CSα/β or ICK motif) and typically have a role in stabilizing peptides by reducing their conformational flexibility. The disulfide bond configuration of chlorotoxin has some parallels to both the cystine-stabilized α/β motif (CSα/β motif) and the inhibitor cystine knot motif (ICK motif). The CSα/β motif is defined by an α-helix and an antiparallel triple-stranded β-sheet connected by two disulfide bonds, CysII-CysV and CysIII-CysVI. In chlorotoxin, Cys5-Cys28, Cys16-Cys33 and Cys20-Cys35 (CysII-CysVI, CysIII-CysVII and CysV-CysVIII) form the CSα/β motif and Cys2-Cys19 (CysI-CysIV) is considered to be an “extra” disulfide bond. The ICK motif incorporates an antiparallel β-sheet stabilized by a cystine knot. The cystine knot is formed by two disulfides (CysI-CysIV, CysII-CysV) that form a ring, with a third disulfide (CysIII-CysVI) penetrating the ring to form the knot. Despite some parallels to both motifs, a recent study showed that the disulfide bond connectivity in chlorotoxin behaves more like a CSα/β motif than an ICK motif. This finding has been reported for other members of the CSα/β family, supporting the conclusion that the configuration of chlorotoxin is more closely related to the CSα/β motif than the ICK motif (Paola G. Ojeda et al., 2015). This is now summarized in the manuscript
20- In the lines 279 to 281 …. Due to its very low concentration in the venom, we couldn't determine the complete amino-acid sequence of this peptide. But with information of a partial peptide reported as above mentioned, line 279, the authors could design a primer of nucleotides and try to clone the gene of targeted peptide from the venom gland! I would like to know the opinion of authors about this possibility?
We agree with the reviewer, we should try to complete the sequence of BotCl by designing a primer of nucleotides and try to clone the gene of BotCl from the venom gland. However, we can have scorpions only in summer and this will take 3 months, not counting the time that takes the orders of the primers.... It can be the subject of future study. This is now added in the manuscript as perspectives (conclusion section).
21- With reference to figure 3, where the authors show the inhibitory activity of the BotCl peptide on the infection of Vero cells caused by the NDV. It would be elegant for them to show an immunofluorescence reaction for virus particles inside the cells. Thus, it would be possible to have a morphological view of the peptide's inhibitory potential and even quantify the inhibition by a second experimental criterion, increasing the competitiveness of the results.
It is indeed interesting to have a fluorescently labeled virus, as it enables the tracking and visualization of virus proliferation within host cells, both in the presence and absence of BotCl. However, in our institution, a market-available fluorescently labeled Newcastle Disease Virus (NDV) is not currently accessible.
To quantify virus inhibition and assess antiviral activity in the presence of BotCl, a standard virus titration method such as plaque assays is employed. Plaque assays are commonly used to determine the viral titer and provide a reference point for evaluating the level of virus inhibition by BotCl. In the plaque assay, the virus is allowed to infect susceptible host cells, and subsequent viral replication leads to the formation of visible plaques, which represent areas of cell death and virus propagation. By quantifying the number of plaques formed, we can determine the viral titer and assess the inhibitory effect of BotCl on virus replication.
While a fluorescently labeled virus would provide additional visual information about virus proliferation, the absence of such a technology does not preclude the evaluation of antiviral activity
22- In the lines 321 to 323, Figure 4, the authors wrote… These results suggested that BotCl has a direct activity virucidal on NDV and may prevent initiation of its growth. In my opinion, the data are clear and show inhibition of viral infection and duplication of viral particles within cells. However, it is not possible to state that the peptides have virucidal action (which implies that it causes the direct death of viruses). This can occur due to molecules present in the serum, proteases, among others, which can destroy virus particles out of cells. The peptide apparently binds to virus and prevents them from entering cells. But even that would have to be shown by fluorescent probes, or radioactive labeling, for example.
We agree with the reviewer that the inactivation of the virus could be due to the binding with BotCl that’s why we omit the “virucidal” term in the manuscript. However, this effect cannot occur due to molecules present in the serum, proteases, among others, which can destroy virus particles out of cells. Indeed cell culture is a widely used method to amplify virus infectivity and determine its titers. In our infection protocol, we utilize a medium containing Fetal bovine serum (FBS), which is a routine procedure that has proven effective in promoting virus replication. It is important to note that FBS, the medium, and the cells themselves do not inhibit virus infection; rather, they enhance virus multiplication. Furthermore, assays used to determine the mode of action of active molecules are well-established and widely used by various research teams around the world (Donalisio, Argenziano et al. 2020, Donalisio, Cirrincione et al. 2020, Civra, A., et al. 2017).
23- About the experiments shown in figure 5, and text between lines 331 to 339, made to show absence of cytotoxicity on bird embryos. It would be interesting to write in the methodology how many times these experiments were repeated. Also, do not stop at the macroscopic analysis of the embryos, but complete with histopathological studies, and finally measure biochemical markers of liver, kidney, heart and blood cytotoxicity at least, if possible, or other possible markers. This would bring more security to the data shown! further into toxicological studies related to the use of BotCl.
The in ovo toxicity assay was assessed three times and we mentioned it in the text.One aspect to explore would be the absence of hemorrhage or any adverse effects caused by BotCl. Hemorrhage is often a concern when assessing the safety of antiviral agents or therapeutic compounds. In our results there is no hemorrhage effect on chicken embryos. The toxicological studies, including in vivo toxicity assays such as the LD50 (lethal dose 50), will be conducted as part of future projects, probably with the recombinant peptide.
24- About the experiments shown in figure 6, and text between lines 342 to 353, made to show protective activity of peptide against NDV infectivity upon bird embryos. It would be interesting to write in the methodology how many times these experiments were repeated. Also, other markers different of macroscopic analysis as pointed in figure 6A. Histopathological analysis of tissues would be useful for complete conclusion. Finally, authors could test greater concentrations of peptide in order to increase the inhibitory effect pointed.
Thank you for your comment. We have made the following revisions in Section 2.9: "The in ovo toxicity assay was repeated three times." Additionally, we have acknowledged the need for further work on histopathology regarding ovo toxicity. I agree with your opinion that increasing the concentration could enhance the inhibition. Unfortunately, we were limited in quantity after conducting the in vitro toxicity assay with the native peptide.
25- In line 346, and figure 6B, please change CN by NC. That is the correct form in English (Negative control).
Done
26- Regarding the data shown in figure 7, they seem clear and well done to me! However, I missed some discussion by the authors about what would be the role of this peptide for scorpions and other animals that produce them? Are scorpions subject to similar virus infections? Or it's an unexpected non-canonical activity? This could be discussed in the text!
Scorpions are not subject to NDV infections. However the Chlorotoxn-like peptides could have broad anti-viral effects. Indeed, scorpion venomous gland contains a wide range of biological active molecules, that are believed to be an integral component of an innate immune system to protect the scorpion and its gland against a variety of pathogen (https://doi.org/10.3390/) . It has been suggested that the presence of AMPs might protect the venom gland from infection and facilitate the action of other neurotoxins (Hernández-Aponte et al., 2011; doi: 10.1016/j.toxicon.2014.06.006). Numerous antimicrobial peptides have now been identified in invertebrates, and they are recognized as playing a key role in protection from pathogenic organisms. Consequently, and given the conservation of glycoproteins type hemagglutinin-neuraminidase (HN) on the surface of different viruses, BotCl could probably block other viral infections. This was discussed in the revised manuscript.
27- About figure 8, although experiments were well done and authors have expertise in the area, they know that these results have a high level of speculation, as the authors do not have co-crystallization data, or molecular dynamic studies, or finally mutated isoforms of toxins that could prove this interaction and inhibit anti viral effects triggered by toxin.
We agree with the reviewer. In the revised version of the manuscript we reported the effect of the native AaCtx, a chlorotoxin-like peptide from Aa scorpion. This peptide showed a significant activity against NDV (Figure 4), even though it showed lesser effect than BotCl. So we believe that the results reported in figure 8 are no longer speculative. Interestingly, we found that the interaction mode of AaCtx is very similar to that of Chlorotoxin (data not shown).
28- Given the conservation of glycoproteins type hemagglutinin-neuraminidase (HN) on the surface of different virus, as discussed in the text (lines 408 to 411) could this peptide block other viral infections?
We thank the reviewer for this comment. In fact BotCl could be active on other viral infections. This is now added to our manuscript (in both discussion and conclusion sections).
29- In lines 422 and 423 … In this study, we screened the venom of Buthus occitanus tunetanus scorpion and isolated an active peptide BotCl. Any special reason other than epidemiological and geographic to study this particular scorpion?
Bot venom is the most represented species in Tunisia and considered as the most toxic after Androctonus australis species. This was added on the manuscript (discussion section)
30- My comments on results shown in figure 8, and text between lines 493 to 512, should not exclude the figure from the text, but only I would like the authors to be more careful in interpreting the data. They should make it clear in the text that these conclusions need to be refined, with the use of techniques such as co-crystallization and site-directed mutations in the peptide.
We agree with the reviewer and the conclusions were modified accordingly.
31- I missed discussions by the authors about more refined work with cloned and recombinant expressed molecules in bacterial models conducive to the synthesis of disulfide bridges, or models with yeast or insect cells (eukaryotes), where mutated forms at important sites of the molecule could also be produced to better study the mechanistic of peptide/NDV interactions.
We agree with the reviewer. Indeed, more refined work with cloned and recombinant expressed molecules in bacterial models conducive to the synthesis of disulfide bridges, where mutated forms at important sites of the molecule could also be produced to better study the mechanistic of peptide/NDV interactions. This is now added to the conclusion as perspectives
32- Alternatively, the authors could obtain synthetic chemical analogues to try to increase the yield in obtaining the peptide of study, and thus better refine the experimental data.
We agree with the reviewer for this suggestion, However we couldn’t buy the synthetic chlorotoxin from Tunisia because of administrative restrictions. That’s why we tested the native AaCtx (Figure 4) an analogue of BotCl, which we previously isolated from Aa scorpion venom (Rjeibi et al., 2011)
33- Finally, the authors tried to study the protective activity of peptide on adult animals exposed to nature conditions and wild type virus? A kind of data that would be very interesting in order to use this peptide as prophylactic or protector against NDV.
We agree with the reviewer that this data would be very interesting, however, BotCl is a native molecule purified from the venom after 3 steps of purification. As a result the obtained quantity is very small and to study adult animals a huge quantity is required, besides that at least 3 animals should be tested. Nevertheless this study could be done with the recombinant peptides as perspective. This is now added to our manuscript (in both discussion and conclusion sections).

Round 2
Reviewer 1 Report
The authors have sufficiently addressed the comments